# A Trajectory of Discovery: Metabolic Regulation by the Conditionally Disordered Chloroplast Protein, CP12

**DOI:** 10.3390/biom12081047

**Published:** 2022-07-28

**Authors:** Cassy Gérard, Frédéric Carrière, Véronique Receveur-Bréchot, Hélène Launay, Brigitte Gontero

**Affiliations:** Aix Marseille Univ, CNRS, BIP, UMR 7281, IMM, FR3479, 31 Chemin J. Aiguier, CEDEX 9, 13 402 Marseille, France; cgerard@imm.cnrs.fr (C.G.); carriere@imm.cnrs.fr (F.C.); veronique.brechot@imm.cnrs.fr (V.R.-B.)

**Keywords:** Calvin-Benson-Bassham cycle, conditionally disordered protein, history of modern science, metabolism regulation, moonlighting protein, protein-protein interaction

## Abstract

The chloroplast protein CP12, which is widespread in photosynthetic organisms, belongs to the intrinsically disordered proteins family. This small protein (80 amino acid residues long) presents a bias in its composition; it is enriched in charged amino acids, has a small number of hydrophobic residues, and has a high proportion of disorder-promoting residues. More precisely, CP12 is a conditionally disordered proteins (CDP) dependent upon the redox state of its four cysteine residues. During the day, reducing conditions prevail in the chloroplast, and CP12 is fully disordered. Under oxidizing conditions (night), its cysteine residues form two disulfide bridges that confer some stability to some structural elements. Like many CDPs, CP12 plays key roles, and its redox-dependent conditional disorder is important for the main function of CP12: the dark/light regulation of the Calvin-Benson-Bassham (CBB) cycle responsible for CO_2_ assimilation. Oxidized CP12 binds to glyceraldehyde-3-phosphate dehydrogenase and phosphoribulokinase and thereby inhibits their activity. However, recent studies reveal that CP12 may have other functions beyond the CBB cycle regulation. In this review, we report the discovery of this protein, its features as a disordered protein, and the many functions this small protein can have.

## 1. Introduction

As V. Uversky mentioned, the discovery of the natural abundance and functional importance of intrinsically disordered proteins (IDPs) has changed protein science [1]. It is now widely accepted that the protein structure-function paradigm that dominated scientific minds for more than 100 years does not hold true for all proteins, and IDPs or proteins containing disordered regions (IDR)s are widespread in all areas of life. IDPs and IDRs differ from structured globular proteins and domains in many respects, such as their amino acid composition, complexity of sequence, hydrophobicity, charge, flexibility, and rate of amino acid substitutions over evolutionary time [2]. They play significant roles in many biological processes, such as control of the cell cycle, transcriptional activation, and signaling, and they frequently interact with many partners or function as central hubs in protein interaction networks. However, in the plant kingdom only a few IDPs have been studied through a recent analysis of 12 plant genomes, which revealed that the occurrence of disorder in plants is similar to that in many other eukaryotes [3]. In plants, most of the information on IDPs comes from *Arabidopsis thaliana*, and among them, to cite but a few, the late embryogenesis abundant (LEA) proteins that are important IDPs are mainly associated with environmental stress [4]. In the algae research field, a recent experimental study reported that 682 proteins from a chlorophyte, *Chlamydomonas reinhardtii*, were heat-resistant, and 299 were predicted to be disordered by four different disorder predictors [5]. However, only a few algal proteins that are fully or partially disordered have been studied [6,7,8,9]. Among them, only the essential pyrenoid component 1 (EPYC1) [10,11,12,13,14], and, above all, the chloroplast protein of 12 kDa (CP12), as evidenced below, have been experimentally studied in depth.

## 2. Discovery of a Small Protein, CP12, in Photosynthetic Organisms

In 1996, cDNA clones were reported from expression libraries for a nuclear-encoded chloroplast protein in three higher plants: pea, spinach, and tobacco, which was named CP12 [15]. This was the first report on CP12, and at this stage, not much was known about this protein. The authors found that this protein consists of about 75 amino acid residues, and it had an abnormal electrophoretic mobility in sodium dodecyl sulfate-polyacrylamide gel electrophoresis (SDS-PAGE) experiments. In addition, they noticed that the CP12 proteins from the three higher plants had a high content of charged amino acid residues, and their conclusion at this time was that this protein was highly hydrophilic and very likely a good candidate for a soluble stroma-located protein within the chloroplast. In all of the three species, two conserved cysteine residues are present at the N- and at the C-terminus that are separated by eight amino acid residues with a central proline residue. The secondary structure prediction suggested that CP12 has a central helix and is organized into two domains, with each containing two cysteine residues that could form disulfide bonds (Figure 1). In this pioneer work, they also showed that CP12 could interact with an enzyme from the Calvin-Benson-Bassham (CBB) cycle, the glyceraldehyde -3-phosphate dehydrogenase (GAPDH, EC 1.2.1.13) [15].

Later, in a work performed on spinach leaves, the same group showed that CP12 not only interacts with GAPDH but also with another enzyme of the CBB cycle, the phosphoribulokinase (PRK, EC 2.7.1.19) [16]. They showed that CP12 could form a 600 kDa complex with the two enzymes mentioned above. This ternary complex was not affected by the presence of NAD(H), but the presence of NADP or NADPH, as well as the high concentration of a reducing agent, dithiothreitol (DTT), led to its dissociation. The authors proposed a model in which the ternary complex exists under dark and dissociates under light. Their results and those of other groups in the literature suggested that the enzymes within the complex were inactive and became active upon dissociation [17,18,19]. Since the CBB cycle does not operate in the dark and become active in the light, the association–dissociation of this complex could be a means to regulate the CBB cycle upon dark-light transitions and reciprocally.

At the same time, Avilan et al. found a complex made up of PRK and GAPDH in the green alga, *C. reinhardtii* [20]. In the green algae, there is a unique form of GAPDH, the homotetrameric A_4_, while in higher plants there is also an A_2_B_2_ where the B-type subunit has a C-terminal extremity that presents homology to the C-terminus of CP12. These authors deeply analyzed the characteristics and the kinetics of the enzymes within the complex and those of the enzymes that dissociated from it [20,21,22,23]. They purified this complex to homogeneity, and, later, the presence of CP12 in the PRK/GAPDH complex reported previously was revealed by MALDI-ToF mass spectrometry [24]. N-terminal sequencing by Edman degradation of the ternary complex allowed one to determine the first amino acid residues of each protein and revealed that the sequence of the mature CP12 starts at SGQPA [25]. Therefore, the complex isolated by Avilan et al. had the same compositions as that found by Wedel et al. Indeed, in 1998, Wedel et al. showed that CP12 was present not only in higher plants but also in *C. reinhardtii*, as well as in many other species (mosses, cyanobacteria). The presence of CP12 in this complex agreed with the cryo-electron microscopy performed on this purified complex, which suggested that other components besides PRK and GAPDH were present [26]. The activities of the enzymes involved in this complex were regulated in vitro by metabolites such as NADP(H) [16,27,28]. All together, these results provided new ideas for the regulation of photosynthesis and were further investigated by many groups.

In *C. reinhardtii*, it was shown that not only the regulatory properties of GAPDH but also its kinetics parameters were affected by CP12. Native GAPDH and recombinant algal GAPDH displayed Michaelis-Menten kinetics with NADH and NADPH as cofactors, with a marked preference for NADPH. Both forms displayed positive cooperativity towards the substrate, 1,3-bisphosphoglycerate (BPGA), but interestingly, these kinetic analyses showed that the native GAPDH had a two-fold lower catalytic constant for the reduction of BPGA, as well as a two-fold lower pseudo-affinity (K_0.5_) for BPGA compared to recombinant GAPDH [24]. These results were surprising, but using mass spectrometry the authors showed that the native GAPDH was still associated with CP12. At the same time, as only a partial sequence of the *C. reinhardtii* CP12 was obtained by PCR, the same authors cloned the entire cDNA of this algal protein and subsequently expressed the protein in *Escherichia coli* [25].

If some results suggested that the PRK/GAPDH/CP12 allowed for the regulation of these enzymes, the presence of this small protein raised a question about its role in the formation of the complex. The role of CP12 in the assembly pathway of the algal PRK/GAPDH/CP12 complex was thus investigated as no complex could be reconstituted in vitro with the native PRK and the recombinant GAPDH devoid of CP12.

In darkened spinach leaves, Scheibe’s group also showed that GAPDH can exist under two inactive aggregated states, one that corresponded to a hexadecameric A_8_B_8_ form and another one that corresponded to the PRK/GAPDH/CP12 complex. Only the dissociation of these edifices with reducing treatment mimicking light resulted in the activity of the released enzymes [29]. The role of CP12 was, by then, far from being understood. Of interest, in the literature, many high oligomerization states of either GAPDH or PRK, in spinach but also in *Phaesolus vulgaris*, have been reported [30]. In spinach, the oligomeric enzymes had latent activity that only appeared upon dissociation [31,32]. In the 20th century, the existence of supramolecular complexes was not recognized in living cells and their existence was seen as artefactual. Therefore, the data were differentially interpreted, but it is very likely that the high molecular mass of GAPDH and PRK with latent activity in fact corresponded to supramolecular complexes.

In the green algae as in the higher plants, it was later shown that the four cysteine residues could form two disulfide bridges, one bridging the N-terminal cysteine pair (residues 23 and 31 in *C. reinhardtii*) and one bridging the C-terminal pair (residues 66 and 75 in *C. reinhardtii*). It was shown using surface plasmon resonance that CP12 under its oxidized state, with two disulfide bridges, was able to bind sequentially to GAPDH with a high affinity (K_D_ equal to 0.44 nM), and then this subcomplex was able to bind to PRK (K_D_ equal to 60 nM). The affinity of CP12 for GAPDH was higher than the one found (µM range) for the Arabidopsis complex [33]. The entity composed of one tetrameric GAPDH, one dimeric PRK, and CP12 (the stoichiometry of which was yet unknown) was defined as a unit. This entity was then able to dimerize to provide the native complex. CP12 therefore acted as a linker in the assembly of the ternary PRK/GAPDH/CP12 complex [25]. Later, native mass spectrometry revealed that two monomeric CP12 molecules were bound to one GAPDH tetramer [34]. Consequently, the stoichiometry inside the ternary complex is two tetrameric GAPDH, two dimeric PRK, and four monomeric CP12. Studies using mutagenesis and limited proteolysis have allowed the residues involved in the interaction between CP12 and GAPDH from *C. reinhardtii* to be mapped and to show that this interaction involves the S-loop arginine residues of GAPDH and the C-terminus of CP12 [35].

## 3. CP12, a Flexible Protein

The first observation of CP12 as an IDP was its abnormal behavior under SDS-PAGE. The protein migrates as a 15 kDa under oxidized form and 25 kDa under its reduced form for *C. reinhardtii*, while the expected theoretical molecular mass of the monomer is 8.5 kDa (Figure 2A,B). Moreover, using size-exclusion chromatography, the elution volume of *C. reinhardtii* CP12 released from the PRK/GAPDH/CP12 complex corresponded to an apparent molecular mass of 35 ± 4 kDa using a column calibrated with globular proteins (Figure 2C). This could correspond to a tetrameric globular form that has never been proven or to an elongated form. These enigmatic behaviors of CP12 were only understood after the concept of IDP was claimed [2,36,37,38]. The size exclusion elution volume mentioned above correlates to a hydrodynamic radius of 2.8 ± 0.1 nm, which corresponds to the expected hydrodynamic properties of a random-coil polymer of 8.8 kDa. These values were in agreement with those confirmed by fluorescence correlation spectroscopy experiments [39]. In 2003, for the first time, it was proposed that the *C. reinhardtii* CP12 belongs to the IDP family (formerly also called an “intrinsically unstructured protein”) [25]. Indeed, CP12 possesses a range of properties that are landmarks of IDPs such as a bias in amino acid composition, is enriched in charged amino acid residues, is depleted in hydrophobic residues, and has a high proportion of disorder-promoting residues (Figure 3). Even if CP12 has a high proportion of disorder-promoting residues, the presence of cysteine residues was first surprising as cysteine residues were considered as “order-promoting residues” due to their ability to form inter- or intramolecular disulfide bridges.

After the identification of CP12 as an IDP, a range of biophysical techniques confirmed that reduced CP12 completely lacks stable secondary and tertiary structural elements. The circular dichroism (CD) spectrum of reduced CP12 (or imitations of reduced CP12 using cysteine to serine mutants) showed a minimum ellipticity at 200 nm, as is characteristic for disordered proteins (Figure 4A) [25]. The Kratky representation of the small-angle X-ray scattering (SAXS) data of reduced CP12 exhibited a plateau at q.Rg > 2 typical of random polymers and characteristic of fully disordered proteins (Figure 4B) [41]. The ^1^H nuclear magnetic resonance (NMR) frequencies of all resonances from reduced CP12 showed minimal chemical shift dispersion (clustered from 7.5 to 8.5 ppm); their linewidths were sharp, and all the features of NMR data were typical of that disordered proteins (Figure 4C) [41]. In addition, NMR data confirmed that reduced CP12 exchanges between a myriad of possible conformations rapidly at a timescale of less than a nanosecond, as expected for an IDP. Other biophysical methods confirmed the IDP properties for reduced CP12, including Förster resonance energy transfer (FRET), fluorescence correlation spectroscopy (FCS), or mass spectrometry [42]. Moreover, it was shown that under oxidized conditions, CP12 was partially folded but still very flexible, and only a model structure obtained by sequence-based molecular modelling was available for many years [43]. CD analysis showed that it was much more helical than in its reduced form (Figure 4A), and an ion-mobility mass spectrometry study showed that the algal oxidized CP12 exists under two conformational states, a compact one and an extended one [34]. Later, experimental data obtained by SAXS also showed atypical features: the Kratky plot of oxidized CP12 was an intermediate between that of a well-folded protein (a bell-shaped curve with a maximum at q.Rg value of √3) and that of a fully disordered protein (such as that of reduced CP12), and these features are characteristic of protein with unstable structural properties (Figure 4B) [44,45]. The SAXS profile revealed the co-existence of two populations of conformers in solution, a compact one and a more disordered one, with all features being characteristic of protein with unstable structural properties. Similarly, the ^1^H-^15^N NMR spectrum of oxidized CP12 differed from that of reduced CP12 and showed a small number of dispersed resonances together with a large number of broad resonances clustered from 7.5 to 8.5 ppm (Figure 4C). All these experimental data could be reconciled with a two-state equilibrium for the algal oxidized CP12: (i) 60% of the oxidized CP12 molecules have two helices in the N-terminal half of the protein and a globular domain at the C-terminus; (ii) 40% of the oxidized CP12 molecules have only the globular fold at the C-terminus, while the N-terminal half remains disordered [44]. The multiple structural transitions and conformational flexibility of CP12 could provide a clue on how this protein can carry variable functions and bind multiple partners. When the stable C-terminal structural element of the *C. reinhardtii* oxidized CP12 binds to GAPDH, it induces a cryptic disorder, and its unstable N-terminal region is further destabilized to favor a disordered conformation [44]. This structural transition upon GAPDH binding contrasts to plant oxidized CP12, where the binding of GAPDH leads to a compaction of the N-terminal region [36,37]. These differences in the stability of the N-terminal region of oxidized CP12 correlate with the differences of relative affinity of CP12 for GAPDH between the algal and the plant species mentioned above with opposite entropic contribution to the binding.

Because the structural properties of CP12 vary significantly depending upon the redox conditions, the term conditionally disordered was coined for this protein. Structural properties of CDP such as CP12 are challenging to analyze [46]. Therefore, the only high-resolution structures available for CP12 are those of oxidized CP12 within the ternary complex and have been deciphered recently by crystallography and cryo-electron microscopy [47,48,49,50].

CP12 is not the unique protein that undergoes structural transitions upon oxidation/reduction, and it is predicted that redox-sensitive CDPs are widespread and have key roles in many eukaryotic processes [51]. Based on the computational platform, IUPred2A, it was predicted that cysteine-rich sequences display significant disorder in the reduced but not the oxidized form, increasing the potential for such sequences to function in a redox-sensitive manner [52]. In photosynthetic organisms where dark-light transitions are correlated to different oxido-reduction conditions, this concept is of paramount importance. The redox structural transitions that have been observed for CP12 might be highly relevant to CP12 being a redox switch of the CBB cycle [53].

## 4. CP12, a Widespread Protein with Sequence Variations on an Original Theme

After 2002, the number of manuscripts dealing with this protein started to increase, and CP12 has been found in many species such as higher plants, microalgae and cyanobacteria [54]. The canonical CP12 sequence contains one N-terminal cysteine residue pair separated by seven or eight residues, one C-terminal cysteine residue pair separated by eight residues encompassing a central proline residue (CxxxPxxxxC), and a core sequence AWD_VEEL (Figure 5). The two pairs of cysteine residues are capable of forming disulfide bridges required to form the ternary complex described above in green algae and higher plants [55]. However, in the glaucophyte *Cyanophora paradoxa*, CP12 lacks the two cysteine residues at the N-terminus [54]. The lack of the N-terminal pair of cysteine residues was also found in the red algal *Galdieria sulphuraria* CP12 and *Synechococcus elongatus* PCC7942, but it did not impair the formation of the ternary complex [56,57]. Though these two cysteine residues were claimed to be important to the PRK binding in higher plants, the presence of the disulfide bond at the N-terminus of CP12 might not be a requisite for PRK binding. It is, however, possible that the affinity between PRK and CP12 is much lower when this disulfide bond is absent and that its absence modulates the stability of the n-terminal helical hairpin described above. Indeed, the mutant of CP12 lacking this disulfide bond is less prone to interact with PRK, but a faint band is still present, indicating a degree of PRK and CP12 interaction [16].

In cyanobacteria, CP12 proteins fused to two cystathionine β-synthase (CBS) domains (CBS-CP12) were found beside the stand-alone CP12, and at present, CBS-fused CP12 has only been reported in these organisms. These CBS-proteins are widespread, and the analysis of 333 cyanobacterial genomes revealed the presence of many variants (Figure 5) [58].

A CP12-like protein was reported in the freshwater diatom, *Asterionella formosa*, that was associated with GAPDH and the ferredoxin NADP reductase, but the sequence of this protein is not available [59,60]. In contrast, in the marine diatom, *Thalassiosira pseudonana*, three CP12 proteins were identified, CP12-1 and CP12-2 were predicted to be localized in the chloroplast, and only CP12-2 was found in expressed sequence tags (ESTs) database and further characterized [61]. The gene coding for this protein in other diatoms was also found. In diatoms, nonetheless, PRK/GAPDH/CP12 has never been found [62], and it seems that the absence of two cysteine residues at positions 245 and 248 on diatom PRK could explain this [63]. Like the canonical CP12, the *T. pseudonana* CP12 possesses some intrinsically disordered regions, is highly dynamic but possesses a central coiled coil motif, and is dimeric, and these characteristics give *T. pseudonana* CP12-2 a form of an elongated cylinder with kinks [61].

The cyanophage-infecting marine picocyanobacteria of the genera Prochlorococcus and Synechococcus have been shown to express a protein that has a C-terminal extension similar to that of CP12. This protein shuts down the CBB cycle, as does the canonical CP12, and uses the NADPH produced by the host to fuel their own deoxynucleotide biosynthesis for replication [64]. Other proteins also possess a C-terminal extension similar to the C-terminus of CP12, such as the B subunit of the higher plant A_2_B_2_ GAPDH, the adenylate kinase 3 (ADK3) from *C. reinhardtii*, and the argininosuccinate lyase. This CxxxPxxxxC extremity interacts with GAPDH in the PRK/GAPDH/CP12, and this interaction is also conserved in the A_2_B_2_ GAPDH and the ADK3 [65,66]. In the prasinophycean green algae, *Ostreococcus tauri* and *Ostreococcus lucimarinus*, CP12 is not present, but they possess the redox-regulated B subunit of GAPDH, which is typical of *Streptophyta* [67].

Three isoforms of CP12 have been reported in higher plants [66]. In *A. thaliana*, the transcripts localization of the isoforms differs; CP12-1 and CP12-2 are mostly expressed in photosynthetic tissues, whereas CP12-3 is expressed in non-photosynthetic tissues such as in the roots. In contrast, in *C. reinhardtii*, one unique isoform has been reported to be localized in the chloroplast. In the C4 plant maize (*Zea mays*), a CP12 homolog was found in the bundle sheath and not in the mesophyll cells [68]. Recently, two CP12 proteins were found in sugarcane, another C4 plant [69]. Though yet never reported, it is very likely that plants with a crassulacean acid metabolism (CAM) also possess this protein. Therefore, it seems that this protein is ubiquitous in the plant kingdom.

## 5. One Gene, One Protein, Many Functions

### 5.1. CP12 Jack-of-All Trades but Master of the CBB Cycle

As mentioned above, CP12 is known to be the master of the CBB cycle [9]. The involvement of oxidized CP12 in supramolecular complexes containing GAPDH and PRK has been demonstrated in several photosynthetic organisms though the strength of binding between these proteins differs among species. As mentioned above, the dissociation constant for GAPDH/CP12 is in the micromolar range in *A. thaliana* [33] but in the nanomolar range in *C. reinhardtii* [25]. The flexibility and the net negative charge of CP12 may increase its reactive area and ‘stickiness’ compared with rigid proteins, thus enhancing the ability of this protein to act as a ‘scaffold protein’ [70].

In *S. elongatus* PCC7942, CP12 forms the ternary complex in response to NADP(H)/NAD(H) ratio. Of interest, most CBB enzymes are not redox-regulated in cyanobacteria [57], whereas in higher plants and green algae, some CBB enzymes are redox-regulated via the thioredoxins (Trx). In Plantae, the Trx participate, in addition to the association-dissociation of the complex PRK/CP12/GAPDH, regulates PRK and GAPDH enzymes activities. PRK and CP12 are reduced by Trx f and m and could be oxidized by the newly characterized TrxLike2 [71,72]. The presence of CP12 has been shown to modify the PRK redox regulation, and, in particular, the formation of the ternary complex decreases the time required for PRK activation [73]. CP12 is also responsible for the redox regulation of the A_4_ form of GAPDH in *C. reinhardtii* [35]. In contrast, the A_2_B_2_ form of GAPDH is autonomously redox-regulated, and CP12 therefore might not be required. Nevertheless, the A_2_B_2_ GAPDH is found in the ternary complex and is more easily activated in dimmer light than the A_8_B_8_ GAPDH oligomer mentioned above [29]. This suggests that CP12 can have other functions than the redox regulation of PRK and GAPDH.

The expression of the genes encoding GAPDH, PRK, and CP12-2 in *A. thaliana* was found to be coordinated, and this suggests that they are regulated at the transcriptional level [74,75]. This suggests that CP12 is involved in the post-translational regulation and at the transcriptional level. A recent study showed that reduced *C. reinhardtii* CP12 stabilizes PRK in vitro and in vivo, but the mechanism of this protection needs further investigation [76,77]. In the mutant strain of *C. reinhardtii*, where the CP12 protein is absent, while the abundance of numerous proteins increases (see below), the abundance of others, including PRK, involved in photosynthesis, decreases. This is in agreement with other studies on *N. tabacum*, *A. thaliana*, and *S. guianensis* that showed that photosynthetic efficiency is reduced in the CP12 deletion mutant [76,77,78,79,80].

### 5.2. CP12, Other Functions

Like many IDPs, CP12 is a promiscuous protein, and in *C. reinhardtii*, in an oxidized state, it can bind other enzymes such as the malate dehydrogenase, the elongation factor 1α2, and 38 kDa ribosome-associated protein, but to a lesser extent than PRK, GAPDH, and the fructose-1,6-bisphosphate aldolase [81]. IDPs are well known to be a hub for the supramolecular complex, but it is surprising that, so far, no interacting partners have been identified for the disordered reduced CP12, and this has probably been overlooked.

Several studies have shown that the role of CP12 is beyond the CBB cycle. In *C. reinhardtii*, the deletion of the protein induced a re-routing of the metabolism under the light. In particular, metabolic pathways involving malate shuttles increased in the mutant such as the tricarboxylic acid cycle (TCA) and the glyoxylate pathway [77]. Malate shuttles, combined with other signaling factors, play a putative role in algal CO_2_-concentrating mechanisms (CCM) [82,83]. In relation to this, it can be noticed that CP12 increases in low CO_2_ conditions in *T. pseudonana*, conditions that trigger CCM [84]. In *N. tabacum* antisense plants, the activity of malate dehydrogenase and glucose-6-phosphate dehydrogenase decreased, and transcripts for polyamine metabolism and polyphenol oxidase were up-regulated [78,79]. A CP12-disrupted strain was engineered in *S. elongatus* PCC7942, and its growth was similar to that of wild-type cells under continuous light but was significantly reduced under the light/dark cycle (12 h/12 h) [57,85]. In the dark, the O_2_ consumption by the mutant strain was lower, and the concentration of ribulose-1,5-bisphosphate, the product of the PRK reaction, was higher than for the wild-type. In cyanobacteria, the main metabolic pathway in the dark is the oxidative pentose phosphate (OPP) pathway that also encompasses the ribulose-1,5 -bisphosphate. By inhibiting the activity of PRK and GAPDH in the dark, CP12 thus regulates the carbon flow from the CBB cycle to the OPP cycle. The authors also found that the cyanobacterial CP12 can bind NADPH (not NADH), but this has not been reported and/or studied to our knowledge in any other CP12. All these results show that the role of CP12 is beyond the regulation of the CBB cycle. In the sugarcane, the expression of one of the isoform of the CP12—ShCP12-1—decreased immediately on the onset of sucrose accumulation that occurs under the yellow canopy syndrome, a specific pattern of leaf yellowing accompanied by abnormal and lethal accumulation of sucrose and starch in leaves [69]. This CP12 might therefore be the primary regulation point of sugar feedback regulation occurring in C4 plants, while the two carboxylating enzymes, ribulose-1,5-bisphosphate carboxylase oxygenase (RuBisCO) and phosphoenolpyruvate carboxylase, were only negatively regulated at a later stage and might be the secondary regulation points.

In cyanobacteria, CP12 has been found in a fusion protein with a CBS domain, as mentioned above. A study of CBS-CP12 from *Microcystis aeruginosa* revealed that the gene expression of this protein is clearly light-induced. In addition, CBS-CP12 oligomerizes and forms a hexamer but does not form the ternary complex with GAPDH and PRK. It can bind AMP and then inhibits the activation of PRK by thioredoxins [86]. The authors propose that this new architecture provides CP12 additional regulatory functions in cyanobacteria.

### 5.3. CP12, an Anti-Stress Protein

In both *A. thaliana* and *N. tabacum*, the antisense suppression of CP12 increased the expression of proteins related to oxidative stress [87]. Recently, it has been shown in *C. reinhardtii* that the suppression of CP12 leads also to an increase in the proteins involved in stress [77]. In addition, in cyanobacteria, CP12 might be involved in oxidative stress by controlling the electrons flux from Photosystem I. Indeed, while the growth at low light of the wild-type and the CP12 mutant strain were the same, at high light the mutant strain grew more slowly. The chlorophyll content also decreased in this strain, and the reactive oxygen species increased [85], while in A. thaliana and *C. reinhardtii*, it has been shown that CP12 provides the thiol groups PRK and GAPDH protection against oxidative damage [87]. In cyanobacteria, the defense mechanism could be different and independent of the thiol groups of these enzymes [57].

In *C. reinhardtii*, CP12 protected GAPDH against heat inactivation and aggregation and therefore plays the role of a specific chaperone [88]. As mentioned above, CP12 also protects PRK against irreversible inactivation in vitro [77]. Besides these roles as a specific chaperone, CP12 from other organisms is more abundant in stress conditions, and this is the case for the *T. pseudonana* CP12-2. The expression of this protein was higher under low CO_2_ [84] but also under N, P, or Si limited conditions [89]. These results therefore indicate that CP12 is not specific to carbon metabolism.

In the tropical legume, *Stylosanthes guianensis*, the higher expression of CP12 increases growth and plant height. In addition to the expected functions, a potential role for CP12 in chilling tolerance has been suggested [80]. A recent transcriptomic analysis of maize also revealed the different regulations of cold-responsive genes and, among them, the CP12 gene is present [90].

All these results show that the role of CP12 is not restricted to the formation of the well-known supramolecular complex involving PRK and GAPDH but is probably more general and characterized not only by conformation heterogeneity but also by functional heterogeneity defining its moonlighting signature as many IDPs.

### 5.4. CP12 and Metal Ions

Metal binding is ubiquitous in biology, being important for folding, stability, transportation, and catalysis [91]. *C. reinhardtii* recombinant CP12 purified by affinity chromatography on nickel columns had a yellow color, even after dialysis with a buffer devoid of metal and imidazole. The absorption spectra from 280 to 600 nm showed the presence of a broad peak around 410 nm, and these spectra strongly resembled those of ferredoxin [92]. Using electrospray non-denaturing mass spectrometry, the authors showed that CP12 was specifically able to bind Cu^2+^ and Ni^2+^ with a low affinity (dissociation constants of 26 and 11 µM, respectively) [92], values close to those obtained for the binding of copper to prion proteins (K_D_ of 14 µM) [93]. Cu^2+^ catalyzed the oxidation of the reduced CP12, with the reformation of disulfide bonds leading to the formation of oxidized CP12, which was able to bind a Cu^2+^ ion. In addition, a hydrophobic cluster analysis showed that CP12 had high similarity with copper chaperones from A. thaliana. Though many questions remain unanswered, one can hypothesize that CP12 may play a role in copper homeostasis like other copper chaperones [94]. Later, using top-down mass spectrometry, three regions were found to be involved in metal ion binding: Asp16-Asp23, Asp38-Lys50, and Asp70-Glu76 [88]. It has been suggested that the binding of copper led to a more rigid structure, but this requires further investigation. Later, using two-dimensional polyacrylamide gel electrophoresis separation of the stroma fraction of *A. thaliana* chloroplasts followed by calcium overlay assay, CP12 was identified as a calcium-binding protein [95]. Though this protein does not possess the canonical calcium-binding EF-hand motif, the authors suggested that negatively charged amino acid residues could be involved in this binding. The biological functions of the Cu^2+^, Ni^2^+, and Ca^2+^ binding, however, remain unsolved and need to be further investigated.

## 6. Conclusions

Photosynthesis regulation depends on many signals, including pH, metabolite concentrations, and oxido-reduction conditions. For photosynthesis to be optimized, the signals received have to be transmitted in a rapid and specific manner and often involve protein-protein interactions; IDPs are well suited for such functions. The chloroplast protein, CP12, a redox dependent conditionally disordered protein, acting as a linker or scaffold between PRK and GAPDH, can integrate these multiple signals to regulate their activity. The redox state of CP12 conditions a severe structural transition of its structural properties from a completely disordered state under reducing conditions to a partially stable state under oxidizing conditions. This redox-dependent structural transition is also concomitant with the association-dissociation with PRK and GAPDH enzymes and thus the regulation of their activity under dark (inactive enzymes) or light (active enzymes). The two enzymes, PRK and GAPDH, do not catalyze consecutive reactions but are using ATP and NADPH, respectively, both products from the primary phase of photosynthesis. PRK produces the RuBisCO substrate, ribulose 1,5-bisphosphate, from the ribulose-5-phosphate, an intermediate of the OPP pathway. GAPDH uses NADPH to produce glyceraldehyde-3-phosphate, which can be exported and is also an intermediate of the OPP. Therefore, CP12 using as a regulatory protein of both PRK and GAPDH, thus “killing two birds with one stone”, contributes to the fine tuning of metabolic pathways such as the CBB cycle, the glycolysis, and the OPP, avoiding futile cycling. It is also involved in the regulation of TCA and glyoxylate cycles involving the malate shuttle and possibly involved in CCM. Moreover, besides its role in controlling metabolic pathways, CP12 provides a cell-signaling pathway, triggers anti-stress responses and protects against oxidative damage. It is also able to bind metal ions, though hitherto the biological significance of this remains unknown (Figure 6). The pursuit of knowledge on these disordered proteins will probably produce new concepts in the sciences as the more we learn and the more questions we will find to ask. The discovery of disordered proteins and of CP12, 70 years after the discovery of the CBB cycle, offers new insights into the photosynthesis field, and this is probably not a dead end.

## Figures and Tables

**Figure 1 biomolecules-12-01047-f001:**
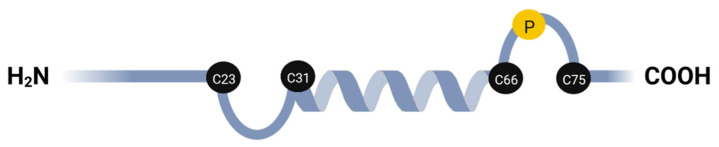
Schematic representation of the predicted secondary structure of mature CP12. Cysteine residues proposed to form peptide loops and that can form two disulfide bridges (C23–C31 and C66–C75) when CP12 is oxidized are indicated with black circles. These two loops are separated by an alpha helix. The proline residue conserved in CP12 from Plantae is shown with a yellow circle. Numbering is from the *C. reinhardtii* mature CP12 sequence. This figure was created with BioRender (https://biorender.com/ (accessed on 25 July 2022)) and adapted from Wedel et al. [16].

**Figure 2 biomolecules-12-01047-f002:**
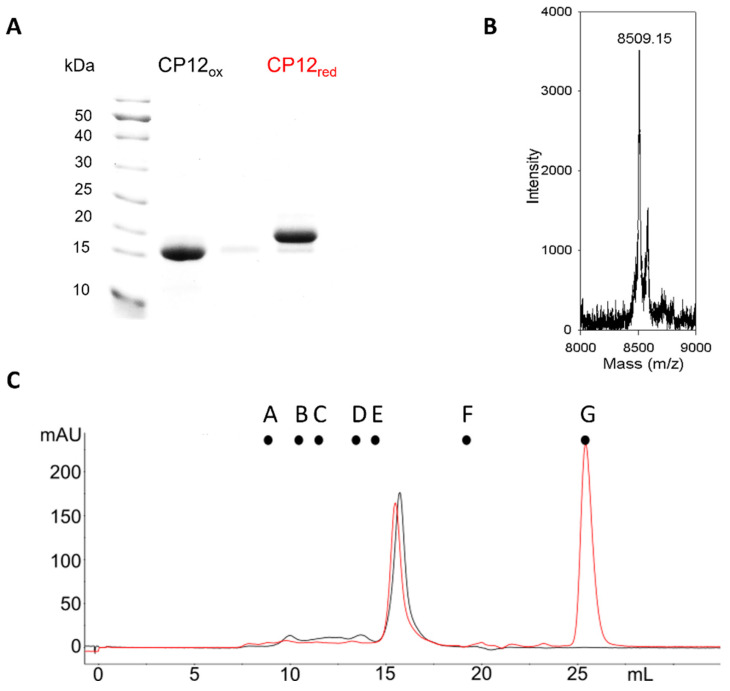
CP12 behaves as an IDP. (**A**) SDS-PAGE (12%) of 4 µg *C. reinhardtii* recombinant CP12 under its oxidized or reduced state. (**B**) MALDI-ToF mass spectrum of the native CP12 isolated from the PRK/GAPDH/CP12 complex of *C. reinhardtii*. (**C**) Size-exclusion chromatography profile of *C. reinhardtii* recombinant CP12 under oxidized (black) or reduced (red) conditions (column: Superdex 200 10 × 300 mm). Above the chromatogram, the dots from A to G indicate the position of molecular-weight standard globular proteins. A: Ferritine (MW 440 kDa, r_H_ 6.8 nm); B: Catalase (MW 240 kDa, r_H_ 5.5 nm); C: dimer of Bovine Serum Albumin (BSA, MW 136 kDa, r_H_ 4.5 nm); D: monomer of BSA (MW 68 kDa, r_H_ 3.5 nm); E: Ovalbumin (MW 43 kDa, r_H_ 3 nm); F: Cytochrome (MW 12.5 kDa, r_H_ 2 nm) C; and G: oxidized form of DTT. MW and r_H_ stand for molecular weight and hydrodynamic radius.

**Figure 3 biomolecules-12-01047-f003:**
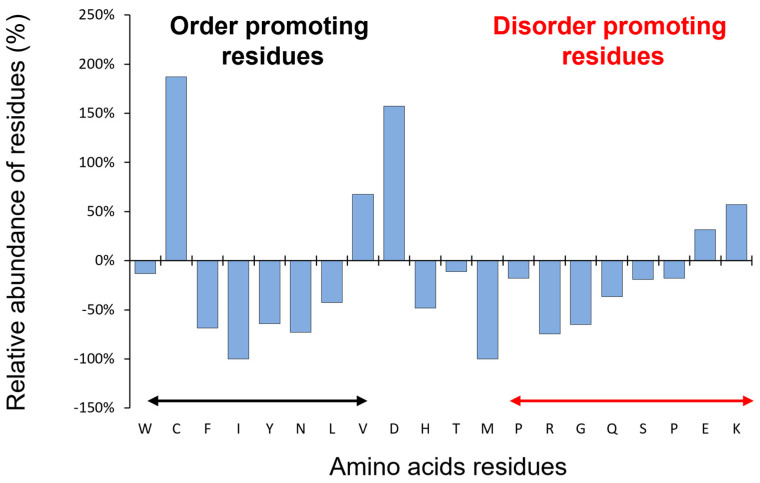
CP12 presents a bias in amino acid composition. Comparison of amino acid composition between globular proteins and *C. reinhardtii* CP12 using composition profiler (http://www.cprofiler.org (accessed on 25 July 2022)) [40]. The globular proteins dataset is from protein data bank (PDB) Select 25.

**Figure 4 biomolecules-12-01047-f004:**
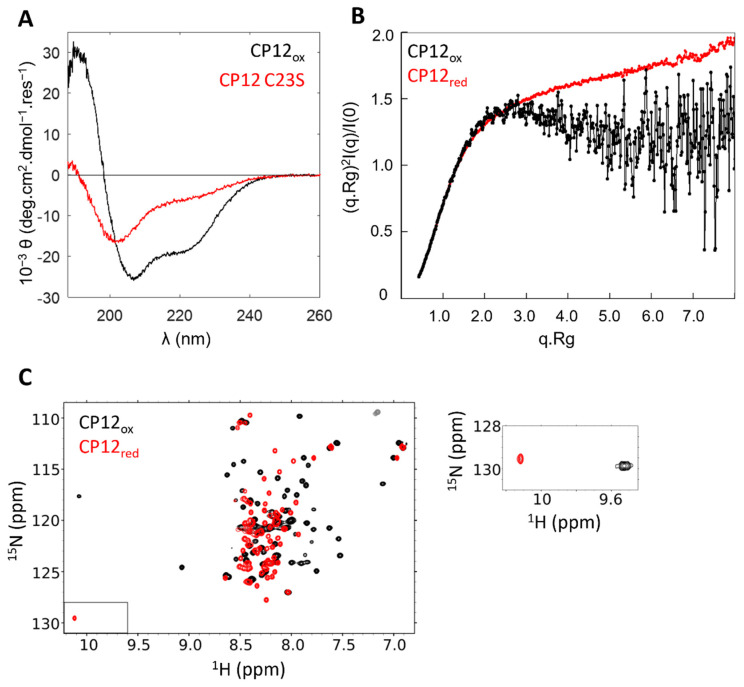
Biophysical analysis of CP12 confirmed that CP12 is an IDP. (**A**) Circular dichroism spectra of 10 µM recombinant *C. reinhardtii* oxidized CP12 (black), and of a CP12 mutant lacking the N-terminal disulfide bridge (mimicking reducing conditions, red). (**B**) Normalised Kratky representation of the SAXS data of the oxidized (black) and reduced (red) form of recombinant *C. reinhardtii* CP12. (**C**) NMR ^1^H-^15^N-HSQC spectra of the oxidized (black) and reduced (red) form of recombinant *C. reinhardtii* CP12. The box between 9.5 and 10 ppm corresponds to the insert shown on the left.

**Figure 5 biomolecules-12-01047-f005:**
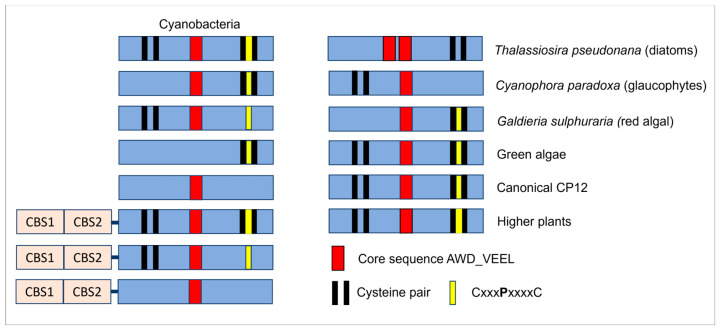
Schematic representation of structural variants of CP12. CBS (orange rectangles) stands for cystathionine β-synthase. The pairs of black lines represent cysteine residues pairs, the red rectangles represent the core sequences AWD_VEEL, and the yellow lines represent the central proline within the C-terminal residues pair. Adapted from D.N Stanley et al., 2013 [58].

**Figure 6 biomolecules-12-01047-f006:**
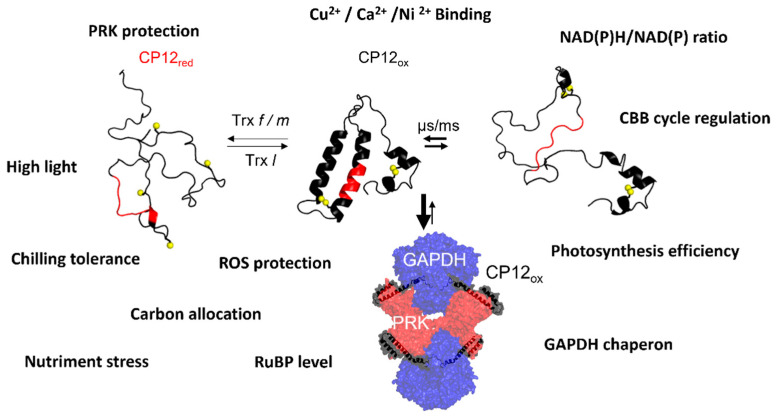
Atlas of CP12 functions. The core sequence of CP12, AWD_VEEL, is in red, and sulfur atoms are indicated in yellow. Reduced CP12 is fully disordered, and oxidized CP12 is partially ordered. Under oxidized state, CP12 forms a supramolecular complex with GAPDH and PRK. A non-exhaustive list of CP12 functions is indicated in the scheme.

## Data Availability

Not applicable.

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
