# Peer review of "A Trajectory of Discovery: Metabolic Regulation by the Conditionally Disordered Chloroplast Protein, CP12"

_biomolecules, 2022, doi:10.3390/biom12081047_

Round 1

Reviewer 1 Report

This paper provides a very good summary of the history of discovery, structure, and function of CP12 in various photosynthetic organisms.  Although the emphasis is on description of structural features and the lack of description of physiological functions is a concern, this manuscript is very worthy for the researchers in this field.  The following points should be considered.

Line 319-329: This is supposed to be a description of CP12 in plants, but it is not clear.  Since cyanobacteria are mentioned at the beginning of the paragraph, it would be better to specify the organisms in each description.

Line 382-383: I could not find the data and description of proteins involved in oxidative stresses in reference 76 and 78.  Please check again.

After page 4, notation of the reference number is not consistent.

Author Response

We thank Reviewer 1 for their comments and have made all their modifications

Line 319-329: This is supposed to be a description of CP12 in plants, but it is not clear.  Since cyanobacteria are mentioned at the beginning of the paragraph, it would be better to specify the organisms in each description.

This has been modified

Line 382-383: I could not find the data and description of proteins involved in oxidative stresses in reference 76 and 78.  Please check again.

We apologized and quoted the right reference in the revised version of the manuscript

After page 4, notation of the reference number is not consistent.

We have carefully checked all the references formatting in the text as well as the list at the end. Modifications have been made accordingly.

Reviewer 2 Report

This review reports the discovery, features and function of a small chloroplast protein, CP12. CP12 is composed of about 80 amino acids enrich in charged amino acids, which is a Conditionally Disordered Proteins (CDPs) dependent on the redox state. In daytime, the stroma is under reducing condition because of photosynthesis, resulting in CP12 is disordered. In night, the oxidized state prevails and the four cysteine residues in CP12 form two disulfide bridges which stabilize the protein. It is found that oxidized CP12 form complexes with glyceraldehyde-3-phosphate dehydrogenase and phosphoribulokinase, the enzymes involved in CBB cycle, thereby the activities of enzymes are inhibited by CP12. Furthermore, CP12 binds other enzymes such as the malate dehydrogenase, the elongation factor, ribosome-associated protein and fructose-1,6-bisphosphate aldolase under oxidized state. In addition to regulate CBB cycle, CP12 might involve in tricarboxylic acid cycle (TCA) and glyoxylate pathways.

CP12 is an important and fascinating protein. This review presents good summarization on the CP12, highlights its known function, and proposes its potential roles in chloroplasts. In line 59, it is better to add a figure here to show the domain and secondary structure information.   

Author Response

CP12 is an important and fascinating protein. This review presents good summarization on the CP12, highlights its known function, and proposes its potential roles in chloroplasts. In line 59, it is better to add a figure here to show the domain and secondary structure information.   

We thank the Reviewer for his/her nice comments and we have now added a new figure as recommended.